# Research on the Adaptability of High-Performance Film for Full Recycling to the Curl-Up Film Collecting Method

**Jie Liu [1], Xuanfeng Liu [2], Yongxin Jiang [2], Xin Zhou [2], Li Zhang [2] and Xuenong Wang [1],\***

1   College of Mechanical and Electrical Engineering, Xinjiang Agricultural University, Urumqi 830052, China; xcx@xjau.edu.cn
2   Institute of Agricultural Mechanization, Xinjiang Academy of Agricultural Sciences, Urumqi 830091, China; njsyyx@xaas.ac.cn (X.L.); jyx@xaas.ac.cn (Y.J.); 171847298@masu.edu.cn (X.Z.); zhangli@xaas.ac.cn (L.Z.)
\*   Correspondence: xjwxn@xaas.ac.cn

**Abstract:** Given the problem of the low tensile performance of the plastic film used in China, which brings about difficulties in curl-up film collecting, in this study, a contrast test was carried out on the tensile property of high-performance film for full recycling and the ordinary polyethylene film (PE film) that is used extensively in China. Test results showed that, within the service period, the elongation at break and tensile yield stress of the high-performance film were higher than those of ordinary polyethylene film, and, within the film-laying period of 0~30 days, the reduction scale of the elongation at break and tensile yield stress was higher than that within the film-laying period of 30~180 days. In this study, in order to obtain the lowest tensile performance of the film by curl-up film collecting, the operation principles of the curl-up film collectors were analyzed. The test on the force of curling up the film in the process of overcoming the force between the film and soil was analyzed. Test and analysis results showed that, for different sampling positions, film pick-up angles, and film types, the tensile stress on the film while pulling it up was within a range of 15.97~21.86 MPa. In order to verify the curling up effect of differently structured film collectors on different types of film with different thicknesses, a field test on film curl-up collecting was designed. A contrast test was carried out on two types of curl-up film collectors, 1JRM-2000 and 11SM-1.2, and the test results showed that the film recycling rate and working performance on the film laid in the same year by the film collector with a fixed film pick-up angle were higher than those for varying film pick-up angles. The curl-up film collector fixed with an automatic film-guiding mechanism is not affected by the velocity difference between the linear velocity of the film curl-up mechanism and the advancing velocity of the machine. The film recycling rate and working performance on the film laid in the same year by the 11SM-1.2 curl-up film collector can meet the operational requirements for collecting high-performance film with thicknesses of 0.008 mm and 0.01 mm. This research can provide a reference for simplifying the structure of residual plastic film collectors, increasing the film recycling rate, and reducing the cost.

**Keywords:** high-performance film for full recycling; film recycling; field experiment; film recycling rate

## 1. Introduction

Film mulching technology has the advantages of increasing temperature and moisture; preventing plant diseases, insects and weeds; and promoting crop growth [1]. In 2019, the amount of plastic film used in China reached $1.379 \times 10^6$ t, and the area covered by plastic film reached $1.76281 \times 10^7$ hm$^2$ [2], which ranked first in the world. However, the farmland residual film recycling technology in China started relatively late, and the long-term, large-scale use of ultra-thin and low-strength plastic film has caused a series of problems, such as soil compaction, a decreased seedling rate, and crop yield reduction [3].

At present, manual recycling is mainly adopted in the treatment of non-point source pollution of farmland residual film, mechanical recycling, and the use of degradable plastic

film. Manual recycling of residual film is time-consuming, laborious, and costly, and it is difficult to motivate farmers [4]. Residual film recycling is not required after laying degradable residual film, since the film can decompose through natural degradation [5], however, degradable plastic film is not yet mature in cost control and production technology; the high cost of use and the unpredictable degradation effect during use make it difficult to implement large-scale promotion and use [6,7]. Mechanical recycling of residual film is currently the most widely used method for its high operating efficiency and low operating cost. The mulch film widely used in China has two levels of thickness, 0.008 mm and 0.01 mm, and its tensile property is lower than the mulch film with a thickness of 0.025 mm or more, which is commonly used abroad. Residual plastic film collectors used abroad are mostly curl-up residual film recycling machines with a simple mechanical structure demanding a good tensile performance of plastic film [8], while development of the residual plastic film collectors used in China is restricted by the poor tensile properties of plastic film. According to the planting mode of crops, a variety of film collectors with different mechanical structures has been developed, mainly including drum type, spring tooth type, and tooth chain type [9], which are not only complex in structure but also have a lower film recycling rate than those developed in foreign countries. Marí et al. [10] studied the application of biodegradable plastic mulch films (BDMs) in strawberry planting, and the research results showed that BDMs are a viable alternative to PE mulch. However, Anunciado [6] pointed out in the study of BDMs that the extent of change to the physicochemical properties of BDMs, due to agricultural weathering, is greatly affected by the polymeric composition and is greater in warmer climates. Steinmetz [11] studied BDMs and mentioned that the high use cost restricted the popularization of BDM. Therefore, due to the high cost of agricultural weathering, the technology of BDMs cannot effectively solve the problem of non-point source pollution of residue film in fields. Zhang et al. [12] performed parameter optimization on the Arc-Shaped Nail-Tooth Roller-Type Recovery Machine for Sowing Layer Residual Film, and the field test results showed that this machine type could achieve a normal residual film collection rate of 66.8% on common polyethylene mulching film. Zhou et al. [13] developed a kind of film collector with a film-removing plate, and this device can achieve a film collection rate of 86.93% on common polyethylene mulching film in ideal conditions. However, in the process of collecting the polyethylene mulching film, there are still residue films uncollected in the field, thus, the film-collecting effect was not satisfactory. Qu et al. [14] replaced the traditional rheological processing of drag and shear on high polymer materials with plasticizing transport based on volume elongational rheology, which reduced the macromolecular chain breakage of high polymer materials and greatly improved the mechanical properties of film molded by processing extreme rheological plastics, such as polyethylene. Based on the complex blow-molding technology, through dynamic distribution, the film can be overlaid for 3–5 layers, and the macromolecules are oriented in different directions between the layers to achieve an interweaving effect; thus, the tensile performance of the film is greatly enhanced, and the "high-performance film for full recycling to the curl-up film recycling method" (which can be called "high-performance film") was developed [15,16]. Since the tensile performance of the high-performance film is better than that of common polyethylene film, laying the high-performance film for full recycling can greatly improve the film collecting rate, and the production cost of the high-performance film is very low compared with BDMs; therefore, this technology has become an effective means to solve non-point source pollution of residue films in agricultural fields.

A contrast test on the tensile property of high-performance film and ordinary polyethylene film under different test factors was carried out, and the variation rules of the tensile properties of both films during the film-laying period of 0–180 days, as well as the minimum tensile level for the 180-day film-laying period, were obtained. Moreover, the operation principles of the curl-up residual plastic film collector were analyzed, and the curl-up collecting of the film for the 180-day film-laying period was carried out. Through an analysis on overcoming the force between the soil and the film during curl-up collecting of the film,

the tensile stresses on the film while the curl-up film collector pulled it up under different test factors were obtained. The field test on the curl-up collecting of film was carried out. By comparing the film recycling rate on the film laid in the same year and the working performance of the two residual plastic film collectors of different structures, the proper structure adaptable to the curl-up collecting of high-performance film was obtained. This research can provide theoretical support for simplifying the structure of residual plastic film collectors, enhancing the film recycling rate, and reducing the cost of film recycling.

## 2. Contrast Test on the Tensile Properties of High-Performance Film and Ordinary Polyethylene Film

In order to obtain the variation law of the tensile properties of the high-performance film and the ordinary polyethylene film laid in a cotton field in Xinjiang within their service period and the minimum tensile level at the end of the service period, the film-laying period, the film thickness, sampling direction, and sampling position were used as test factors; the elongation at break and tensile yield stress were used as test indexes to carry out the contrast test on the two types of films.

### 2.1. Basic Information of the Test Field

Maigaiti County is located in the southwestern part of Xinjiang Uygur Autonomous Region, which includes the western part of the Tarim Basin, the eastern part of the Kashgar region, the southwestern edge of the Taklimakan Desert, the northern foot of the Karakoram Mountains, the lower reaches of the Yarkant River, and the lower reaches of the Tiznafu River (77°28′–79°05′ east longitude, 38°25′–39°22′ north latitude). This county has a temperate continental dry climate with sufficient sunshine, a large temperature difference between day and night, very little precipitation, hot summers and cold winters, and a windy and sandy spring. The average annual sunshine is 2836.5 h, the annual average temperature is 11.8 °C, and the annual average precipitation is 56.5 mm.

### 2.2. Test Materials and Field Management

Considering local production conditions, the high-performance film and ordinary polyethylene film with thicknesses of 0.008 mm and 0.01 mm were laid on the cotton test field in Maigaiti county on 30 April 2021. The film-laying site is shown in Figure 1. The planting mode of one film, which covered three pipes and six rows with 660 mm + 100 mm of machine-harvested cotton was adopted in the test field. The plant spacing was 12.5 cm, and routine management of the field was adopted for water–fertilizer management. The high-performance film was manufactured by Guangdong Siico Technology Co., Ltd., (Guangdong, China); the ordinary polyethylene film is manufactured by Xingnong Industry and Trade Co., Ltd. in Bayingolin Mongol Autonomous Prefecture, in Xinjiang province, China. The film-laying situation in the test field is shown in Figure 1.

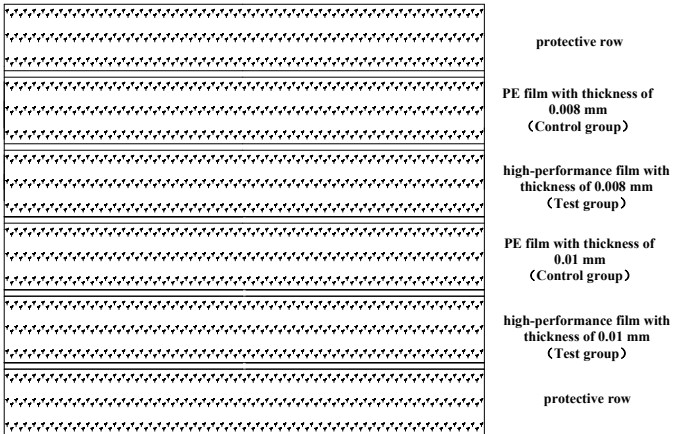

**Figure 1.** Diagram of plastic film laying in test field.

*2.3. Test Design*

2.3.1. Test Factors and Levels

Both high-performance film and ordinary polyethylene film are made from high-molecular compounds [17]. Therefore, at the same sampling spot, their tensile performance is affected mainly by natural erosion, material aging, material thickness, and material anisotropy [18]. Thus, the film-laying period, sampling position, film thickness, and sampling direction were chosen as the test factors in the test on the film tensile property.

Material anisotropy determines that different tensile properties are obtained by testing the film from different directions. Hence, the direction along the film-laying direction was defined as the horizontal direction, while the perpendicular direction of the film-laying direction was defined as the vertical direction. The degree of sunniness and the natural erosion effect on the different positions of the film (near and far away from the plants) may vary due to degree of shading of the cotton plants on the film, thus, the sample-taking positions on the film were divided into near-end positions and far-end positions.

2.3.2. Test Indexes

According to the requirements of GB/T 1040.3-2006 Plastics—Determination of Tensile Properties, the elongation at break of the film and the tensile yield stress were taken as the test indexes, and the calculation method is as follows:

$$\varepsilon_t = \frac{L - L_0}{L_0} \times 100\% \tag{1}$$

where $L$ is the distance between the marked lines when the sample is torn off, mm; $L_0$ is the distance between the original graticule lines, mm.

$$\sigma_t = \frac{F_b}{bd} \tag{2}$$

where $F_b$ is the breaking load of the sample, N; $b$ is the sample width, mm; and $d$ is the sample thickness, mm.

2.3.3. Determination of Test Parameters

The strain data sample frequency is obtained based on test speed, the ratio of the distance between the original graticule lines of the standard sample and the original clamp distance, and the minimum resolution of the obtained strain signal of the accurate data, and its calculation method is as follows:

$$f_{\min} = \frac{vL_0}{60L_c r} \tag{3}$$

where $f_{\min}$ is the sampling frequency of minimum strain data, Hz; $v$ is the test speed, mm/min; $L_c$ is original clamp distance, mm; and $r$ is the minimum resolution of the obtained strain signal of the accurate data, mm.

According to the recommended test speed and the original clamp distance of the standard samples in GB/T 1040.1-2018, $v$ = 10 mm/min, $L_c$ = 115 mm, the CMT-6103 electronic universal testing machine, which is controlled by a microcomputer, obtained the minimum resolution of the obtained strain signal of the accurate data, which was 0.008 mm. After calculation, the sampling frequency of the minimum strain data was obtained, and $f_{\min}$ = 9.06 Hz.

The load data sampling frequency is based on the test speed, strain range, minimum resolution of the obtained strain signal of accurate data, and the initial clamp distance, in which the elastic modulus, test speed, and clamp distance determine the load growth rate. The ratio between the load growth rate and the minimum resolution of the obtained strain

signal of accurate data determines the load data sampling frequency of the test machine. The calculation method is as follows:

$$f_{force} = \frac{\dot{F}}{r} = \frac{v}{\Delta\varepsilon \times 60 \times L_c \times 5 \times 10^{-3}}\tag{4}$$

where $\dot{F}$ is the load growth rate, %, and $\Delta\varepsilon$ is the strain range of the samples. $\Delta\varepsilon = 3 \times 10^{-2}$ was selected according to standard requirements, and the sampling frequency of the load data was calculated to be 9.66 Hz.

In this test, an extensometer is used as the strain indicating device, and it should be a Level 1 extensometer as required by GB/T 12160-2019, that is, the relative error of the gauge length is ±1%, the percent of reading is 0.5%, the absolute value is 1 μm, the relative error is ±1%, and the absolute error is ±3 μm.

In order to avoid the toe at the initial stage in the stress–strain curve, in measuring the related stress, the prestress on the sample before the test should satisfy Equation (5) as follows:

$$0 < \sigma_0 \le \sigma^*/100\tag{5}$$

where $\sigma_0$ is the prestress at the beginning of the test, MPa; $\sigma^*$ is the tensile yield stress of the material, MPa. In order to make the prestress at the beginning of the test adapt to the two types of film, $\sigma^*$ should be less than the lower value of the tensile yield stress of the two types of film; thus, $\sigma_0 = 0.09$ Mpa was selected [19].

### 2.3.4. Sample Collection

The service period of the film laid on the cotton field of south Xinjiang in China is about 180 d. In order to reflect the tensile property variation process of the two types of film during their service periods, film samples were collected every 30 d from the film-laying date to carry out the tensile property test; the samples were collected seven times. Each time, the sampling objects included two sets of high-performance film and ordinary polyethylene film of 0.008 mm and 0.01 mm in thickness, with a width of slightly more than 300 mm and a length of slightly more than 660 mm. After sample collection, the film samples were rinsed to remove the impurities for airing. On each selected sample film, eight standard tensile pieces were cut down by a cutter and used as test material, as shown in Figure 2. The size of the standard tensile film pieces is shown in Figure 3. During each instance of sample collection, the intact film sample pieces were obtained on dry, hard, flat land, and the sampling positions were marked on the film.

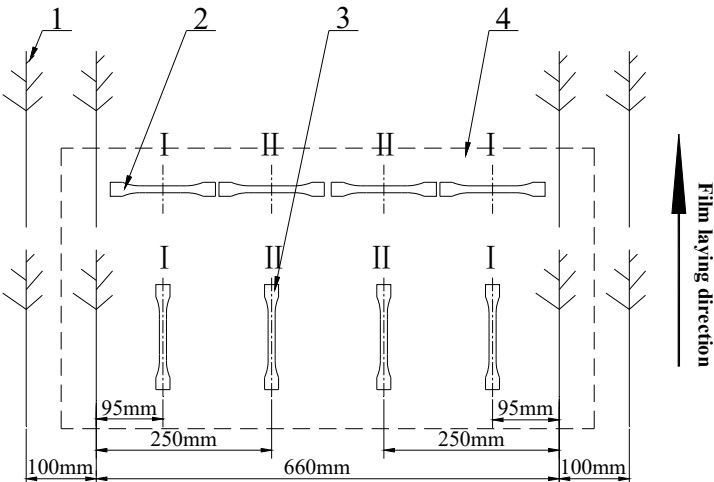

**Figure 2.** Schematic diagram of sampling location: 1—cotton plant, 2—vertical film sampling, 3—horizontal film sampling, 4—film sample piece, I—near-end position, II—far-end position.

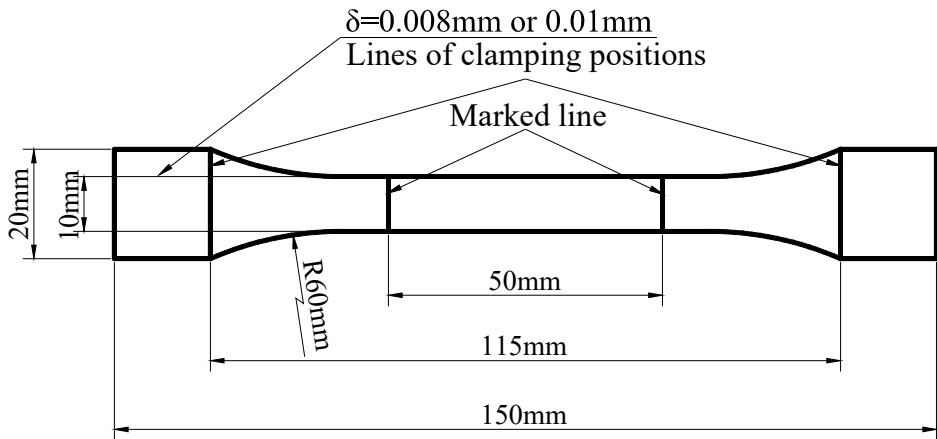

**Figure 3.** Standard tensile sample of film.

2.3.5. Test Scheme

Before the test, a low-power magnifying glass was used to check the test samples; the sample pieces with unsmooth and frayed edges or damages were eliminated to avoid test errors caused by stress concentration on the damaged parts of the sample pieces in the test. The CMT-6103 electronic universal testing machine controlled by a microcomputer was used to carry out a test on the film tensile property. According to Equations (1) and (2), the elongation at break and tensile yield stress of the film were calculated. The test was repeated four times, and test results were averaged. The test process is shown in Figure 4. Figure 4a shows the state of the sample after prestressing, and Figure 4b–d show the tensile process of the sample after loading.

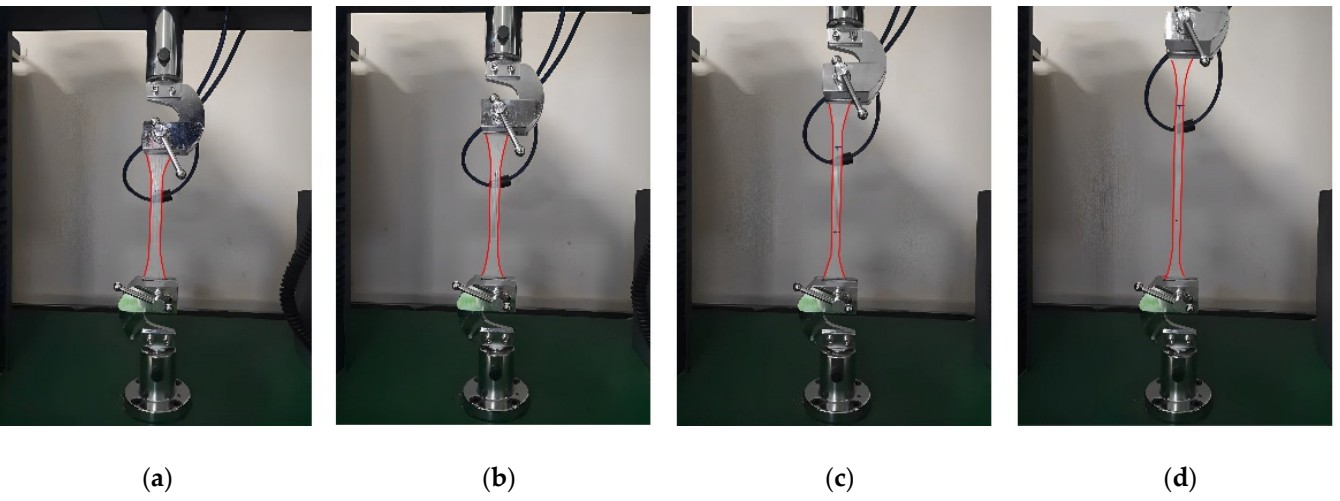

| (a) | (b) | (c) | (d) |

**Figure 4.** Process of the tensile test of film. (**a**) shows the state of the sample after prestressing, (**b**–**d**) show the tensile process of the sample after loading.

## 3. Test on Curl-Up Force in Film Collecting

The curl-up residual plastic film collector is generally composed of the film pick-up mechanism, film-guiding mechanism, film-curling mechanism, impurity separation mechanism, and film-unloading mechanism [20]. During operation, the film pick-up mechanism loosens the soil on the film surface on both sides of the film and separates the film from the soil [21]. Then, the film-guiding mechanism transmits the film to the impurity separation mechanism to the film-curling mechanism. The impurity separation mechanism separates the soil, roots, and stems from the film through vibration or sweeping. The film-curling mechanism curls up the film to a suitable size, and, finally, the film-unloading device unloads the residue film package after curling up.

In the test on the curl-up force during film collecting, by simulating the process of overcoming the force from the soil to the film during curl-up collecting of the residue film, the tensile stresses on the film while the curl-up film collector pulls up the film under different test factors were obtained. In collecting film, the film pick-up mechanism separates the film from the soil and forms a film pick-up angle $\alpha$; the curl-up force $F$ is formed in curl-up collecting film. The force between the film and soil under the effect of the curl-up force is shown in Figure 5. Since the soil on the film's surface at the slope has the tendency to move downwards, there is a friction $f_2$ from the film against the soil on the film at the slope. At the same time, the film is uncovered by the film pick-up mechanism along the film pick-up angle $\alpha$. The cohesion force between the film and soil prevents the film from moving and forms a downward force $F_a$ along the film pick-up angle $\alpha$.

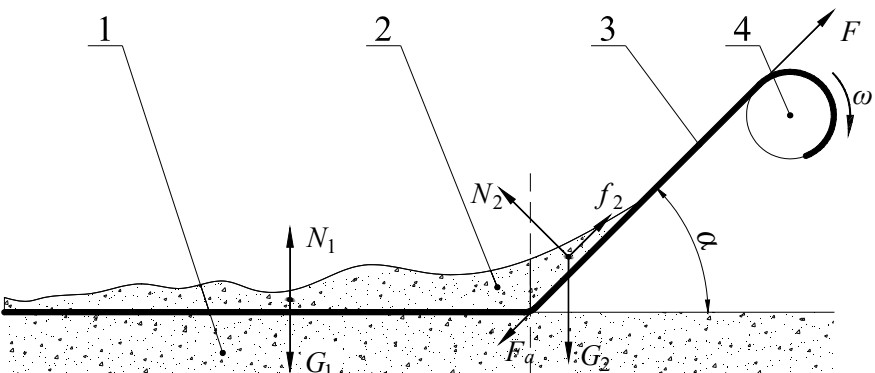

**Figure 5.** Diagram of force between plastic film and soil under the action of curl-up force: 1—soil under the film, 2—soil on the film, 3—film, 4—film-curling mechanism.

In Figure 5, $N_1$ is the support force from the soil and film on the flat ground to the soil on the film; $G_1$ is the gravity of the soil on the film; $N_2$ is the support force from the film at the slope to the soil on the film; and $G_2$ is the gravity of soil on the film at the slope. Then, the mechanics equilibrium equation during operation of the curl-up residual plastic film collector is established as follows:

$$\begin{cases} F = F_a + G_2 \sin \alpha - f_2 \\ N_2 - G_2 \cos \alpha = 0 \\ N_1 - G_1 = 0 \end{cases} \tag{6}$$

In order to prevent the film from being torn down due to the speed difference between the linear velocity of the film-curling mechanism and the advancing speed of the machine, the linear velocity of the curling speed should be equal to the advancing speed of the machine, and the speed should be uniform, so as to avoid tearing down the film with the rigid impact from an abrupt change in the film collecting speed. The test on the curl-up force in film collecting was carried out. By measuring the curl-up force $F$, the tensile stresses on film during the curl-up collecting process under different factor levels were obtained.

### 3.1. Test Conditions

The field test was carried out at the field research and development base of the Northwest Oasis Agricultural Environment Key Laboratory of the Ministry of Agriculture, Tuobuliqi Town, Korla City, Bayingolin Mongolian Autonomous Prefecture, Xinjiang Uygur Autonomous Region in early November 2021. The planting mode (660 mm (wide row) + 100 mm (narrow row)) with protective rows on both sides was adopted, and the film thicknesses were 0.008 mm and 0.01 mm for both the high-performance film and ordinary polyethylene film. The ground was relatively flat, and the drip irrigation belt had been recycled. Using the TZS-1 soil moisture tester, the moisture content of the surface soil was

16.2%. Before the test, the height of the stubbles in the test field was controlled within 120 mm. The test field is shown in Figure 6.

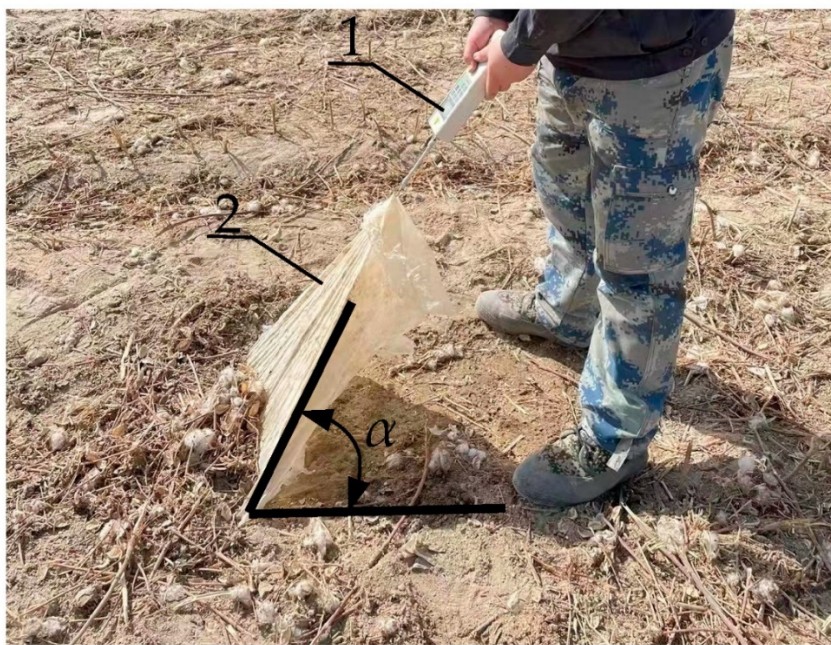

**Figure 6.** Test field on curl-up force in film collecting: 1—HP-50 type Digital Display Pull and Push Strength Calculator, 2—film.

*3.2. Test Method and Design*

3.2.1. Test Factors and Levels

It can be known based on Equation (6) that the value of the curl-up force $F$ is related to cohesion between the soil under the film and the film $F_a$, the film pick-up angle, the gravity of the soil on the film at the slope in the film pick-up $G_2$, and the friction $f_2$ between the film and the soil on the film. Since the moisture content of the soil under the film of different types is different, the higher the moisture content under the film, the higher the cohesion Fa of the soil under the film to the film. The mass of soil on the film is related to the film-laying position. Since cotton plants can shield sandstorms, with the passage of time, the mass of soil near the middle part of the field is lower, and the friction of the film to the soil on the film at the slope is related to the friction coefficient between the soil friction and soil, as well as the mass of soil on the film. Therefore, the sampling position, film pick-up angle, and the types and positions of the laid film were used as test factors. For each planting line of 100 m, the front point of each line was defined as position 1, and 25 m from position 1 along the film-laying direction was defined as position 2; 50 m from position 1 along the film-laying direction was defined as sampling position 3. According to the film pick-up angle of the 1JRM-2000 curl-up film collector, the standard range of the film pick-up angle was determined to be 30–75°. The table of test factor levels in the test on the curl-up force during film collecting is shown in Table 1.

**Table 1.** Test factor levels.

| Levels | Sampling Position | Film Pick-Up Angle | Type of Film |
|:---:|:---:|:---:|:---:|
| 1 | Position 1 | 30° | High-performance film |
| 2 | Position 2 | 45° | Ordinary polyethylene film |
| 3 | Position 3 | 60° | |
| 4 | | 75° | |

### 3.2.2. Test Method

The tensile stress on the film was selected as the test index, which is calculated by Equation (7):

$$\sigma = \frac{F}{bd} \tag{7}$$

where $\sigma$ is the tensile stress on the film, MPa.

In the test, the process of generating the curl-up force on the film with the curl-up film collector was simulated. Figure 7 shows the diagram of the operation process of the 1JRM-2000 curl-up film collector.

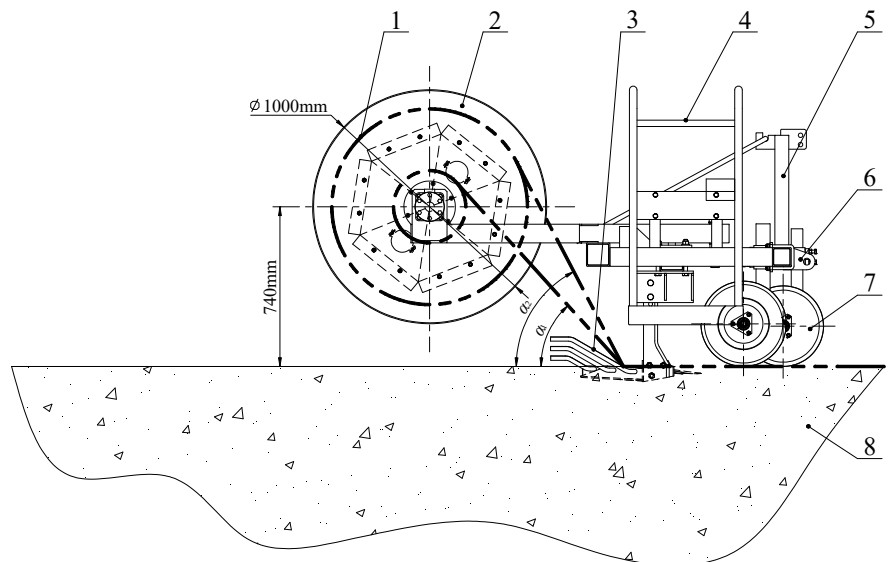

**Figure 7.** Operation process of the 1JRM-2000 curl-up film collector: 1—film, 2—film-curling mechanism, 3—film pick-up mechanism, 4—operation platform, 5—body frame, 6—traction mechanism, 7—deep limiter, 8—soil.

During operation, the variation range of the film pick-up angle is $\alpha_1$-$\alpha_2$. According to Figure 8, during the operation process of the curl-up film collector, the collected residue film would continually wrap around the film-curling device, increasing the film pick-up angle with the increase in the diameter of the residue film wrapping around the film-curling device. The HP-50 digital display pull- and push-strength calculator was adopted to measure the curl-up force. During the force measurement, one end of the film was connected with the pull and push strength calculator, and the other end was at different angles with the ground to simulate the changing process of film pick-up angle during the curl-up collecting of film. The value of the film pick-up angle is controlled by the digital display angle ruler. When the film is initially pulled up, the soil on the film accumulates, and the film is subject to greater soil gravity. When the film is pulled up higher, the accumulation speed of the soil is similar to that of soil falling down from the film. At this time, the soil gravity is in dynamic equilibrium, and the curl-up force becomes stable. The digital display pull- and push-strength calculator was used to record the maximum value of the curl-up force in pulling up the film, and the obtained curl-up force was substituted into Equation (7) to calculate the tensile stress of the film.

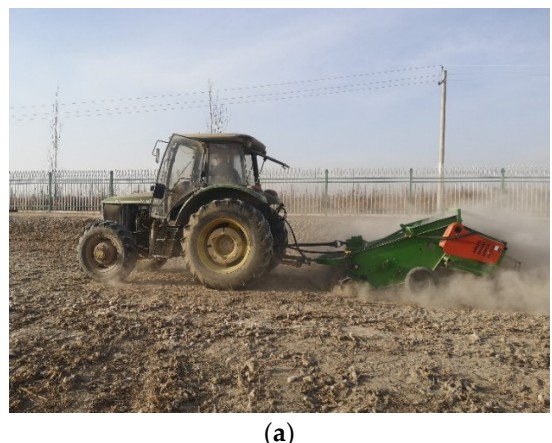
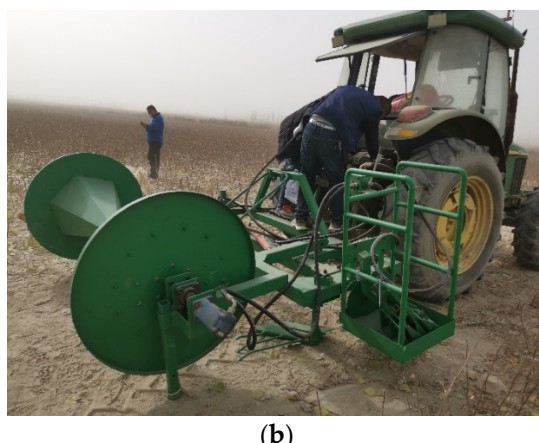

(**a**)　　　　　　　　　　　　　　　　　　　　　　　　(**b**)

**Figure 8.** Test field of curl-up collecting of film. (**a**) 11SM-1.2 curl-up film collector; (**b**) 1JRM-2000 curl-up film collector.

### 3.3. Results and Analysis

3.3.1. Results and Analysis of Contrast Test on the Tensile Properties of High-Performance Film and Ordinary Polyethylene Film

Table 2 shows the contrast test results of the tensile properties of the high-performance film and the ordinary polyethylene film laid in the Xinjiang cotton fields with a service period of 0–180 days.

Table 2 shows that the elongation at break and tensile yield stress of the high-performance film before and during use were higher than those of the ordinary polyethylene film; the elongation at break and tensile yield stress of the film with a thickness of 0.01 mm were higher than those of the film with a thickness of 0.008 mm. The tensile property of the film at a near-end position was higher than that of the film at a far-end position. When the sampling direction was horizontal, the elongation at break and tensile yield stress of the ordinary polyethylene film were higher than those when the film was collected vertically. For the high-performance film, and the elongation at break collected horizontally was higher than that collected vertically; its tensile yield stress was lower than that collected vertically. This is due to the different anisotropy of the high-performance film from the ordinary polyethylene film caused by the orientation of the macromolecules between the layers of the high-performance film. With the increase in the film-laying period, both the elongation at break and tensile yield stress of the high-performance film and ordinary polyethylene film decreased. The variation in the scales of the decrease in the elongation at break and tensile yield stress of the film is shown in Table 3. During the film-laying period of 0~30 days, the scales of the decrease in the elongation at break and tensile yield stress were higher than those during the film-laying period of 30~180 days. When the film-laying period was 120 days and 180 days, the scale of decrease in the elongation at break of the ordinary polyethylene film with a thickness of 0.01 mm collected horizontally at a far-end position and the high-performance film with a thickness of 0.008 mm collected horizontally at a near-end position were negative. This is caused by difference in the thickness of the film and different sampling positions, since the thickness error of film is +0.003~−0.002 mm. Each instance of sampling is located at that of the previous instance; thus, it may have little effect on the scale of decrease in the elongation at break of the film, which shows that there was little variation in the tensile property of the film when the film-laying periods were 90~120 days and 150~180 days.

**Table 2.** Results of film tensile test during service period of 0–180 d.

| Samples | 0.008 mm Horizontal | | | | 0.008 mm Vertical | | | | 0.01 mm Horizontal | | | | 0.01 mm Vertical | | | |
|---|---|---|---|---|---|---|---|---|---|---|---|---|---|---|---|---|
| | Ordinary Polyethylene Film | | High-Performance Film | | Ordinary Polyethylene Film | | High-Performance Film | | Ordinary Polyethylene Film | | High-Performance Film | | Ordinary Polyethylene Film | | High-Performance Film | |
| Days of Film laying/Day | Elongation at Break/% | Tensile Yield Stress/MPa | Elongation at Break/% | Tensile Yield Stress/MPa | Elongation at Break/% | Tensile Yield Stress/MPa | Elongation at Break/% | Tensile Yield Stress/MPa | Elongation at Break/% | Tensile Yield Stress/MPa | Elongation at Break/% | Tensile Yield Stress/MPa | Elongation at Break/% | Tensile Yield Stress/MPa | Elongation at Break/% | Tensile Yield Stress/MPa |
| 0 | 286.715 | 22.37 | 504.052 | 29.814 | 270.146 | 19.738 | 400.837 | 39.313 | 331.216 | 26.773 | 823.628 | 34.55 | 297.147 | 24.957 | 554.794 | 41.35 |

| Samples | 0.008 mm Horizontal near-end position | | | | 0.008 mm Horizontal far-end position | | | | 0.008 mm Vertical near-end position | | | | 0.008 mm Vertical far-end position | | | | 0.01 mm Horizontal near-end position | | | | 0.01 mm Horizontal far-end position | | | | 0.01 mm Vertical near-end position | | | | 0.01 mm Vertical far-end position | | | |
|---|---|---|---|---|---|---|---|---|---|---|---|---|---|---|---|---|---|---|---|---|---|---|---|---|---|---|---|---|---|---|---|---|---|
| | Ordinary polyethylene film | | High-performance film | | Ordinary polyethylene film | | High-performance film | | Ordinary polyethylene film | | High-performance film | | Ordinary polyethylene film | | High-performance film | | Ordinary polyethylene film | | High-performance film | | Ordinary polyethylene film | | High-performance film | | Ordinary polyethylene film | | High-performance film | | Ordinary polyethylene film | | High-performance film | |
| Days of film laying/day | Elongation at break/% | Tensile yield stress/MPa | Elongation at break/% | Tensile yield stress/MPa | Elongation at break/% | Tensile yield stress/MPa | Elongation at break/% | Tensile yield stress/MPa | Elongation at break/% | Tensile yield stress/MPa | Elongation at break/% | Tensile yield stress/MPa | Elongation at break/% | Tensile yield stress/MPa | Elongation at break/% | Tensile yield stress/MPa | Elongation at break/% | Tensile yield stress/MPa | Elongation at break/% | Tensile yield stress/MPa | Elongation at break/% | Tensile yield stress/MPa | Elongation at break/% | Tensile yield stress/MPa | Elongation at break/% | Tensile yield stress/MPa | Elongation at break/% | Tensile yield stress/MPa | Elongation at break/% | Tensile yield stress/MPa | Elongation at break/% | Tensile yield stress/MPa |
| 30 | 246.164 | 19.16 | 481.141 | 25.738 | 217.43 | 18.671 | 467.345 | 24.681 | 223.514 | 17.211 | 371.159 | 33.746 | 211.214 | 15.71 | 366.024 | 32.12 | 289.416 | 23.114 | 781.976 | 29.875 | 263.41 | 22.371 | 740.591 | 29.617 | 260.179 | 21.011 | 513.261 | 36.351 | 243.633 | 20.843 | 463.48 | 35.56 |
| 60 | 231.313 | 17.13 | 461.457 | 24.363 | 186.431 | 16.241 | 440.531 | 22.1 | 209.131 | 15.678 | 356.166 | 30.925 | 193.864 | 14.133 | 342.711 | 28.167 | 266.147 | 20.416 | 751.88 | 27.2 | 239.172 | 19.997 | 702.467 | 27.542 | 237.638 | 19.371 | 487.531 | 34.172 | 216.137 | 18.361 | 415.167 | 32.3 |
| 90 | 226.173 | 16.734 | 455.63 | 24.025 | 177.214 | 15.716 | 433.131 | 21.437 | 197.216 | 15.13 | 347.01 | 30.173 | 179.317 | 13.326 | 331.463 | 26.857 | 251.147 | 19.824 | 746.02 | 26.613 | 220.491 | 18.625 | 681.016 | 25.971 | 218.697 | 18.019 | 469.837 | 33.713 | 197.083 | 16.91 | 406.014 | 31.163 |
| 120 | 213.163 | 16.037 | 450.57 | 23.71 | 163.517 | 14.971 | 423.53 | 20.973 | 188.314 | 14.316 | 339.751 | 29.613 | 165.214 | 12.1 | 322.3 | 26.382 | 243.732 | 18.863 | 739.515 | 26.17 | 223.863 | 17.173 | 663.389 | 24.663 | 208.136 | 17.164 | 460.173 | 33.164 | 181.691 | 16.052 | 398.845 | 29.622 |
| 150 | 201.21 | 15.616 | 447.214 | 23.338 | 149.73 | 14.1 | 417.502 | 20.313 | 178.132 | 13.18 | 329.383 | 29.088 | 147.612 | 10.937 | 313.918 | 25.626 | 237.281 | 18.014 | 728.165 | 25.841 | 221.316 | 16.538 | 644.807 | 23.546 | 201.066 | 16.62 | 447.136 | 32.721 | 174.25 | 15.166 | 390.649 | 28.65 |
| 180 | 191.147 | 15.183 | 450.307 | 22.961 | 136.246 | 13.17 | 412.137 | 19.782 | 170.214 | 12.873 | 321.088 | 28.617 | 131.371 | 9.864 | 305.866 | 24.791 | 226.391 | 17.631 | 717.243 | 25.17 | 204.851 | 16.031 | 628.324 | 22.927 | 196.213 | 16.033 | 438.217 | 32.467 | 166.261 | 14.527 | 383.439 | 28.038 |

**Table 3.** Variation of scale of decrease in film tensile properties with film-laying period.

| Samples | 0.008 mm Horizontal Near-End Position | | | | 0.008 mm Horizontal Far-end Position | | | | 0.008 mm Vertical Near-end Position | | | | 0.008 mm Vertical Far-End Position | | | |
|---|---|---|---|---|---|---|---|---|---|---|---|---|---|---|---|---|
| | Ordinary Polyethylene Film | | High-Performance Film | | Ordinary Polyethylene Film | | High-Performance Film | | Ordinary Polyethylene Film | | High-Performance Film | | Ordinary Polyethylene Film | | High-Performance Film | |
| Days of Film Lay-ing/Day | Scale of De-crease in Elon-gation at Break/% | Scale of De-crease in Tensile Yield Stress/% | Scale of De-crease in Elon-gation at Break/% | Scale of De-crease in Tensile Yield Stress/% | Scale of De-crease in Elon-gation at Break/% | Scale of De-crease in Tensile Yield Stress/% | Scale of De-crease in Elon-gation at Break/% | Scale of De-crease in Tensile Yield Stress/% | Scale of De-crease in Elon-gation at Break/% | Scale of De-crease in Tensile Yield Stress/% | Scale of De-crease in Elon-gation at Break/% | Scale of De-crease in Tensile Yield Stress/% | Scale of De-crease in Elon-gation at Break/% | Scale of De-crease in Tensile Yield Stress/% | Scale of De-crease in Elon-gation at Break/% | Scale of De-crease in Tensile Yield Stress/% |
| 30 | 14.14331 | 14.34958 | 4.54536 | 13.67143 | 24.16511 | 16.53554 | 7.28238 | 17.21674 | 17.26178 | 12.80272 | 7.40401 | 14.16071 | 21.81487 | 20.40734 | 8.68508 | 18.29675 |
| 60 | 6.03297 | 10.59499 | 4.09111 | 5.3423 | 14.257 | 13.01484 | 5.73752 | 10.45744 | 6.43494 | 8.90709 | 4.03951 | 8.35951 | 8.21442 | 10.03819 | 6.36925 | 12.30697 |
| 90 | 2.2221 | 2.31173 | 1.26274 | 1.38735 | 4.94392 | 3.23256 | 1.67979 | 3 | 5.69739 | 3.49534 | 2.57071 | 2.43169 | 7.50371 | 5.71004 | 3.28207 | 4.65083 |
| 120 | 5.75223 | 4.16517 | 1.11055 | 1.31113 | 7.72907 | 4.74039 | 2.21665 | 2.16448 | 4.51383 | 5.38004 | 2.09187 | 1.85596 | 7.86484 | 9.20006 | 2.76441 | 1.76863 |
| 150 | 5.60745 | 2.62518 | 0.74483 | 1.56896 | 8.43154 | 5.81791 | 1.42328 | 3.1469 | 5.40693 | 7.93518 | 3.05165 | 1.77287 | 10.65406 | 9.61157 | 2.60068 | 2.86559 |
| 180 | 5.00124 | 2.7728 | −0.69162 | 1.61539 | 9.00554 | 6.59574 | 1.28502 | 2.61409 | 4.44502 | 2.32929 | 2.51834 | 1.61922 | 11.00249 | 9.81073 | 2.565 | 3.25841 |

| Samples | 0.01 mm Horizontal near-end position | | | | 0.01 mm Horizontal far-end position | | | | 0.01 mm Vertical near-end position | | | | 0.01 mm Vertical far-end position | | | |
|---|---|---|---|---|---|---|---|---|---|---|---|---|---|---|---|---|
| | Ordinary polyethylene film | | High-performance film | | Ordinary polyethylene film | | High-performance film | | Ordinary polyethylene film | | High-performance film | | Ordinary polyethylene film | | High-performance film | |
| Days of film lay-ing/day | Scale of de-crease in elon-gation at break/% | Scale of de-crease in tensile yield stress/% | Scale of de-crease in elon-gation at break/% | Scale of de-crease in tensile yield stress/% | Scale of de-crease in elon-gation at break/% | Scale of de-crease in tensile yield stress/% | Scale of de-crease in elon-gation at break/% | Scale of de-crease in tensile yield stress/% | Scale of de-crease in elon-gation at break/% | Scale of de-crease in tensile yield stress/% | Scale of de-crease in elon-gation at break/% | Scale of de-crease in tensile yield stress/% | Scale of de-crease in elon-gation at break/% | Scale of de-crease in tensile yield stress/% | Scale of de-crease in elon-gation at break/% | Scale of de-crease in tensile yield stress/% |
| 30 | 12.62016 | 13.66675 | 5.05714 | 13.53111 | 20.47184 | 16.44194 | 10.08186 | 14.27786 | 12.44098 | 15.8112 | 7.4862 | 13.75203 | 18.00927 | 16.48435 | 16.45908 | 14.00242 |
| 60 | 8.03998 | 11.67258 | 3.84871 | 8.95397 | 9.20162 | 10.61195 | 5.14778 | 7.00611 | 8.66365 | 7.80544 | 5.01304 | 5.99433 | 11.28583 | 11.90807 | 10.42397 | 9.1676 |
| 90 | 5.63598 | 2.89969 | 0.77938 | 2.15809 | 7.8107 | 6.86103 | 3.05367 | 5.70402 | 7.97053 | 6.97951 | 3.62931 | 1.3432 | 8.8157 | 7.90262 | 2.20465 | 3.52941 |
| 120 | 2.95245 | 4.84766 | 0.87196 | 1.6646 | −1.52931 | 7.79597 | 2.58834 | 5.03639 | 4.82906 | 4.74499 | 2.05688 | 1.62845 | 7.80991 | 5.07392 | 1.7657 | 4.94223 |
| 150 | 2.64676 | 4.50087 | 1.53479 | 1.25716 | 4.13535 | 3.69766 | 2.80107 | 4.52905 | 3.39682 | 3.16942 | 2.83306 | 1.33579 | 4.09541 | 5.51956 | 2.05493 | 3.27481 |
| 180 | 4.5895 | 2.12612 | 1.49993 | 2.59665 | 7.43959 | 3.06567 | 2.55627 | 2.6289 | 2.41364 | 3.53189 | 1.9947 | 0.77626 | 4.58479 | 4.21337 | 1.84565 | 2.16405 |

3.3.2. Results and Analysis of Test on Curl-Up Force in Film Collecting

The software Allpairs was used to generate a hybrid orthogonal table for the test [22], and the test results are shown in Table 4.

**Table 4.** Test plans and results.

| Test No. | Sampling Position | Film Pick-Up Angle/° | Type of Film | The Tensile Stress on the Film/MPa |
|---|---|---|---|---|
| 1 | Position 1 | 30 | High-performance film | 21.86 |
| 2 | Position 2 | 30 | Ordinary polyethylene film | 19.125 |
| 3 | Position 1 | 45 | Ordinary polyethylene film | 19.364 |
| 4 | Position 2 | 45 | High-performance film | 19.83 |
| 5 | Position 3 | 60 | High-performance film | 16.427 |
| 6 | Position 1 | 60 | Ordinary polyethylene film | 18.217 |
| 7 | Position 3 | 75 | Ordinary polyethylene film | 15.97 |
| 8 | Position 1 | 75 | High-performance film | 17.039 |
| 9 | Position 3 | 30 | High-performance film | 17.513 |
| 10 | Position 3 | 45 | Ordinary polyethylene film | 16.824 |
| 11 | Position 2 | 60 | High-performance film | 17.726 |
| 12 | Position 2 | 75 | Ordinary polyethylene film | 16.013 |
| $(k_1)_1$ | 19.12 | 19.499 | 18.399 | |
| $(k_1)_2$ | 18.174 | 18.673 | 17.586 | |
| $(k_1)_3$ | 16.684 | 17.457 | | |
| $(k_1)_4$ | | 16.341 | | |
| $R_1$ | 2.436 | 3.158 | 0.813 | |

According to the analysis of the results in Table 4, it can be obtained that under different test factors, the required film tensile stress for the operation of the curl-up film collector was 15.97–21.86 MPa. By comparing the value with the results of the film tensile property test, the minimum tensile yield stress of the high-performance film with a thickness of 0.01 mm was higher than the required minimum film tensile stress during normal operation of the curl-up film collector. The results of the range analysis showed that the influence order of the test factors on the film tensile stress was *Film Pick-up Angle > Sampling Position > Type of Film*; the film tensile stress achieved the maximum value when position 1 was chosen as the sampling position, the film pick-up angle was 30°, and the film type was high-performance film.

In order to find out the significance level of the test factors on the test indexes, a variance analysis was made on the above test results, and the analysis results are shown in Table 5.

**Table 5.** Variance analysis.

| Indexes | Sources of Variance | Sum of Squares | Degree of Freedom | Mean Square | F Value | Significance |
|---|---|---|---|---|---|---|
| The tensile stress *Y* on the film/MPa | *Sampling Position* | 12.07 | 2 | 6.035 | 6.771 | ** |
| | *Film Pick-up Angle* | 16.07 | 3 | 5.357 | 6.01 | ** |
| | *Type of Film* | 0.81 | 1 | 0.81 | 0.909 | |
| | Residual error | 4.457 | 5 | 0.891 | | |
| | Sum | 33.407 | 11 | | | |

Note: ** means the effect is very significant.

According to the analysis results in Table 4, the required film tensile stress for the operation of the curl-up film collector under different test factors was 15.97~21.86 MPa. By comparing this range with the results of the film tensile property test, only the minimum tensile yield stress of the high-performance film with a thickness of 0.01 mm was higher than the minimum film tensile stress required in normal operation of the curl-up film collector. The range analysis results showed that the influence order of the test factors on the film tensile stress was Film Pick-up Angle > *Sampling Position > Type of Film*; the

film tensile stress achieved the maximum value when position 1 was used as the sampling position, the film pick-up angle was 30°, and the film type was high-performance film.

In order to verify the significance level of each test factor on the test indexes, a variance analysis was made on the above test results, and the analysis results are shown in Table 5.

It can be observed from Table 5 that the sampling position and film pick-up angle had significant influence on the film tensile stress, while the type of film had an insignificant influence on the film tensile stress. During the service period of the film, due to various reasons, such as the wind-blown sand, the soil on the film accumulates. Since the cotton plants can stop the sand, the soil accumulated around the center of each row along the film-laying direction decreases; the longer the service period of the film, the more obvious this tendency becomes. Therefore, when the sampling position was the front point of each row, the soil quantity on the film was highest; thus, the curl-up force required to pull up the film is very high. With the shift of the sampling position to the center of each row and, therefore, with less soil on the film, the curl-up force required to pull up the film reduces. According to Equation (7), the film tensile stress is directly proportional to the curl-up force; thus, the sampling position had a significant influence on the film tensile stress. The angle between the direction of the curl-up force and the ground is equal to the film pick-up angle. The larger the film pick-up angle, the larger the valid component force to pull up the film would become, and the smaller the curl-up force is required. Thus, the film pick-up angle had a significant influence on the film tensile stress. Although the type of film has influence on the soil's moisture content under the film, it has small influence on the cohesion of the soil under the film and the gravity of the soil on the film; thus, the type of film has an insignificant influence on the film tensile stress.

## 4. Field Test on Curl-Up Collecting of Film

By considering the test results of the contrast test on the tensile property under different test factors and the field test on film curl-up collecting between high-performance film and ordinary polyethylene film, the high-performance film with a thickness of 0.01 mm satisfied the requirements for the tensile stress of film in curl-up collecting. Since the value of the film pick-up angle is inversely proportional to the required curl-up force during curl-up collecting of the film, the film pick-up angle of the machine was set to 45°–75° for film collecting. In order to verify the effect of the curl-up collecting of the film for film collectors with different structures on different types of film with different thicknesses, a test on field film curl-up collecting was designed, and the test site is shown in Figure 8.

### 4.1. Test Method and Design

The 1JRM-2000 curl-up film collector and the 11SM-1.2 curl-up film collector were used for a contrast test in the field research and development base of the Key Laboratory of Northwest Oasis Agricultural Environment of Ministry of Agriculture, in Tuobuliqi Town, Korla City, Bayingolin Mongolian Autonomous Prefecture of Xinjiang Uygur Autonomous Region, China, during March of 2022. According to the standard GB/T25412-2021, the film recycling rate of the device on the film laid in the same year and the working performance of the device were used as test indexes. The structures of the two types of collectors are shown in Figure 9.

According to Figure 9, when the 11SM-1.2 curl-up film collector was working, the eight groups of film pick-up mechanisms at the front and the two groups of side-film shovels separated the film and soil; the film-guiding and impurity separation mechanisms separated impurities from the film and sent the film to the film-curling mechanism. The film-curling mechanism rotated and winded the film on it. While unloading the film, the hydrocylinder was manually controlled, and the film unloading mechanism unloaded the film package. During the working process, the film pick-up angle remained unchanged and was determined by the angle of the film pick-up mechanism. If the film pick-up angle is too large, the soil penetration angle of the film pick-up mechanism is too large, and the soil produces high resistance against the film pick-up mechanism. If the film pick-up

angle is too small, it produces high film tensile stress and tears off the film. Thus, the film pick-up angle was determined to be 45°. During operation of the 1JRM-2000 curl-up film collector, the soil-loosening shovel on the deep limiter in the front of the film collector first loosens the soil around the side film. The film-cutting mechanism cuts the soil from the center along the film-laying direction, and then the film pick-up mechanism in the middle of the machine separates the cut film from the soil. With the forward movement of the machine, by manually controlling the hydrocylinder, the film-unloading mechanism opens, and, during the working process of the machine, the film pick-up angle increases with the increase in the diameter of the film package.

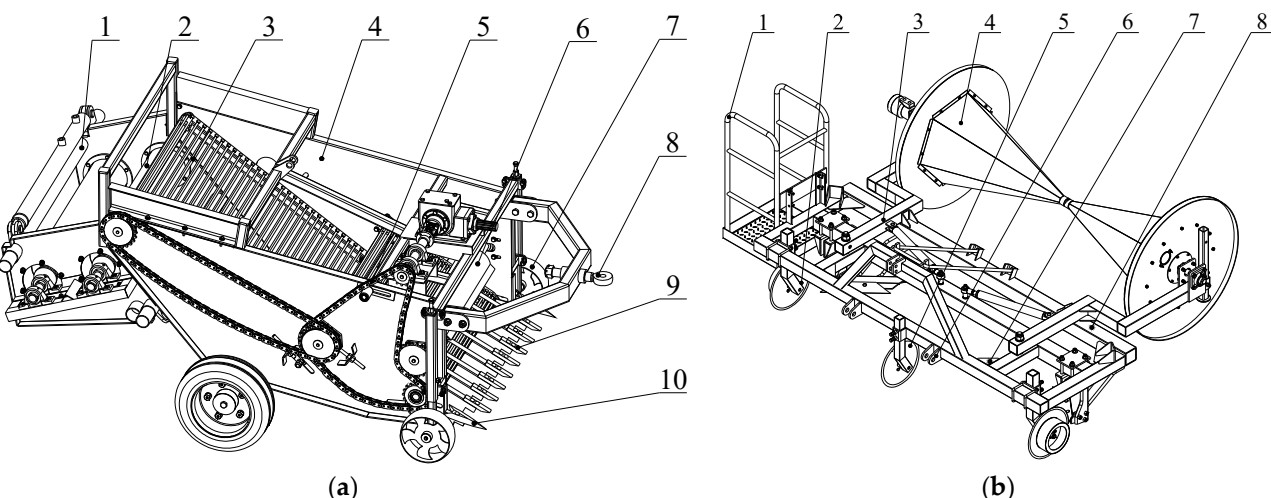

(**a**)　　　　　　　　　　　　　　　　　　　　(**b**)

**Figure 9.** Schematic diagram of structure of two types of curl-up film collectors. (**a**) 11SM-1.2 curl-up film collector: 1—film-unloading mechanism, 2—film-curling mechanism, 3—film-guiding and impurity separation mechanism, 4—body frame, 5—drive system, 6—film-pulling mechanism, 7—depth wheel components, 8—traction mechanism, 9—film pick-up mechanism, 10—side-film shovel; (**b**) 1JRM-2000 curl-up film collector: 1—operation platform, 2—depth limiter, 3—film-unloading mechanism, 3—film-curling mechanism, 4—film-cutting mechanism, 5—traction mechanism, 6—film pick-up mechanism, 8—body frame.

According to the requirements of "five-point random sampling" [23], a measurement area of 200 m × 3.8 m was selected, and test points were chosen within this area. From the four corners of the measurement area along the diagonal lines, four measurement points were randomly determined within the range of one-quarter to one-eighth of the diagonal length, in addition to the intersection of the diagonal lines as the five pre-operation measuring points. Then, five points were selected as post-operation measuring points in the same area near to, but not overlapping, the five pre-operation measuring points. The measuring points cover a length of 5 m and the width of the film, which is 1.25 m. After controlling the stubble height of the cotton plants within 120 mm, the two film collectors started working simultaneously from the start of each row on the same type of film for an operation length of 200 m. The test was repeated three times, and the test results were averaged. Before the machine reached the operation position, a length of 50 was set as the accelerating region to let the machine adjust to a suitable speed. Timing started when the machine entered the operation position, and timing stopped after the machine completed an operation length of 200 m. After operation, residue film pieces were taken from the five pre-operation and post-operation measurement points in the two measurement areas. The residual film taken from each measuring point was washed, dried, and weighed, and the average value was calculated. The film recycling rate on the film laid in the same year can be calculated according to Equation (8):

$$J = (1 - \frac{W}{W_0}) \times 100\% \qquad (8)$$

where *J* is the film recycling rate of the film laid in the same year by the machine, %; *W* is the mass of residue film laid in the same year in the field after machine operation, g; and $W_0$ is the mass of residue film laid in the same year before machine operation, g.

### 4.2. Test Results and Analysis

Test results are listed in Table 6. The results of the test come from "five-point sampling", which is suitable for the survey objects with relatively uniform population distribution and good representativeness. In order to exclude errors caused by accidental factors, three replicate groups were arranged for each sampling, and the final results were averaged.

**Table 6.** Results of field test on curl-up collecting of film.

| Test No. | Type of Collector | Type of Film | Thickness of Film/mm | Film Recycling Rate of the Film Laid in the Same Year/% | Working Performance/km·h |
|---|---|---|---|---|---|
| 1 | 1JRM-2000 | High-performance film | 0.01 | 81.16 | 6.15 |
| 2 | 1JRM-2000 | High-performance film | 0.008 | 73.26 | 4.79 |
| 3 | 1JRM-2000 | Ordinary polyethylene film | 0.01 | 57.31 | 3.76 |
| 4 | 1JRM-2000 | Ordinary polyethylene film | 0.008 | 55.43 | 3.13 |
| 5 | 11SM-1.2 | High-performance film | 0.01 | 96.11 | 9.37 |
| 6 | 11SM-1.2 | High-performance film | 0.008 | 85.45 | 8.24 |
| 7 | 11SM-1.2 | Ordinary polyethylene film | 0.01 | 78.52 | 7.88 |
| 8 | 11SM-1.2 | Ordinary polyethylene film | 0.008 | 72.49 | 7.64 |

Table 6 shows that, during the curl-up film collecting of the 1JRM-2000 curl-up film collector on film with different thicknesses, the film recycling rate of the film laid in the same year and the working performance were lower than that of the 11SM-1.2 curl-up film collector. During the working process of the 1JRM-2000 curl-up film collector, with the increase in the film pick-up angle, the curl-up force changes, and the film is easily broken down during film pick-up. In order to collect the film more easily, the 1JRM-2000 film collector used soil-loosening shovels to loosen the soil around the side film to reduce the force on film. After the soil was loosened, some side film still adhered to the soil and could not be collected, making the film recycling rate of this device lower than that of the 11SM-1.2 curl-up film collector. When the type of film to be collected was high-performance film, since the mechanical properties of the high-performance film were higher than those of the ordinary polyethylene film, the film-cutting mechanism could not effectively cut off the high-performance film, thereby preventing the machine from improving the working performance. When the type of film to be collected was high-performance film, since the mechanical properties of the high-performance film were higher than those of the ordinary polyethylene film, the film-cutting mechanism could not effectively cut it off, which shows the low working performance of the machine. When the type of film to be collected was ordinary polyethylene film, whose minimum tensile yield stress should be lower than the required film tensile stress for the normal operation of the curl-up film collector, the force direction on the film kept changing during operation, and the film was easily broken. In this case, it was necessary to pull the broken film manually to the film-curling mechanism, and thus the working performance of the machine was greatly affected. Since the film pick-up angle of the 11SM-1.2 curl-up film collector is a fixed value, during collecting of the film, the curl-up force is only determined by factors such as the soil quantity on the film. When there is little change in the curl-up force, the film is not broken, and, moreover, with the assistance of the film-guiding mechanism, in the case of film breakage during curl-up collecting, the film-guiding mechanism can transmit the newly separated film from the soil to the film-curling mechanism without manual operation. It can be obtained from the results of the field test on the curl-up collecting of the film that the 11SM-1.2 curl-up film collector achieved film recycling rates of 85.45% and 96.11% on the high-performance film with thicknesses of 0.008~0.01 mm laid in the same year; the 1JRM-2000 curl-up film collector achieved the film recycling rate of 81.16% on the high-performance film

with a thickness of 0.01 mm laid in the same year, which could satisfy the requirements of GB/T25412-2021 and achieved working performances of 8.24 km/h, 9.37 km/h, and 6.15 km/h and satisfied the requirements for agricultural production.

Due to the long-term use of ultra-thin and low-strength plastic films in China, the residual film collectors developed in China are mainly aimed at collecting low-tensile strength plastic films. The current related researches includes: The Agricultural Mechanization Research Institute of Xinjiang Academy of Agricultural Sciences [24] has developed a 4JSM-2.1A arc-reciprocating residual film collector; Jiangsu University [25] has developed a combined residual film reclaimer with upper conveyor chain; and China Agricultural University [26] has developed a collecting and separating device for strip plastic film baler. The residual film collected by this device is fragmented, and the film collection mechanism also collects some impurities into the film collecting box during the recycling process, so the collected film can only be reused through granulation, and it is difficult to completely remove impurities, such as the straw, soil and other impurities mixed in the residual film fragments. The cost of using residual film for granulation remains high, and many downstream enterprises of residual film recycling should only rely on government subsidies to support them. It can be concluded in this study that the tensile strength and weather resistance of the high-performance film for full recycling are better than those of the ordinary polyethylene film, and the residual film can be recycled by means of pick-up recycling. The collection of low-tensile strength plastic film and the collected plastic film with high integrity have relatively few impurities, which greatly reduces the cost of collecting residual film for downstream enterprises.

## 5. Conclusions

(1) A contrast test was carried out on the tensile properties of high-performance film and ordinary polyethylene film, and the test results showed that the elongation at break and the yield stress of the high-performance film before and during the operation were higher than those of the ordinary polyethylene film. The tensile property at a near-end position of the cotton plants was higher than that for a far-end position. When the sampling direction was horizontal, the elongation at break and the tensile yield stress of the ordinary polyethylene film were higher than those when the sampling direction was vertical, and the elongation at break of the high-performance film was higher than that when the sampling direction was vertical, its tensile yield stress was lower than that when the sampling direction was vertical. With the increase in the film laying period, the elongation at break and tensile yield stress had downward tendencies, and, within 0–30 days, the scales of decrease in the elongation at break and tensile yield stress were higher than those during 30–180 days.

(2) Test results showed that the range in tensile stress on the film was 15.97~21.86 MPa when the film is pulled up from different sampling positions, at different film pick-up angles, and with different types of film. The minimum tensile yield stress of the high-performance film with a thickness of 0.01 mm was higher than the maximum film tensile stress required for pulling up the film by the curl-up film collector. The influence order of the test factors on the film tensile stress was film pick-up angle > sampling position > type of film. After a variance analysis on the test data, the results showed that the sampling position and film pick-up angle had significant influences on the tensile stress of the film, while the type of film had an insignificant influence.

(3) Test results showed that during operation of the film collectors, the 11SM-1.2 curl-up film collector with a fixed film pick-up angle achieved a higher film recycling rate on the film laid in the same year and a higher working performance in collecting film of different types and with different thicknesses than the 1JRM-2000 curl-up film collector. The 11SM-1.2 curl-up film collector achieved a film recycling rate of 85.45% and 96.11% on the high-performance film with thicknesses of 0.008 mm and 0.01 mm. The 1JRM-2000 curl-up film collector achieved a film recycling rate of 81.16% on the high-performance film with a thickness of 0.01 mm laid in the same year, which satisfied the requirements of

GB/T25412-2021. Its working performances were 8.24 km/h, 9.37 km/h, and 6.15 km/h, respectively, which could satisfy the demand in production.

(4) In real production, the linear velocity of the film-curling mechanism and the advancing speed of the machine cannot be equally consistent; therefore, the monitoring-feedback–control system is generally adopted to realize a dynamic equilibrium between the linear velocity of the film-curling mechanism and the advancing speed of the machine, thus enhancing the complexity of the machine. If the difference between the linear velocity of the film-curling mechanism and the advancing speed of the machine is too large, the film is easily torn off. Since the automatic film-guiding mechanism can automatically supply film, the working performance of the 11SM-1.2 curl-up film collector is not affected by the difference between the linear velocity of the film-curling mechanism and the advancing speed of the machine.

(5) In the future, we can optimize the curl-up collecting method of film collectors from the perspective of a simulation analysis, and subsequent tests should consider test indexes, such as the number of instances of film breakage and the impurity rate of the film, to find out the optimal mechanical structure and working parameters, and to make preparations for secondary or multiple utilizations of the collected film.

**Author Contributions:** Conceptualization, methodology, investigation, data curation, formal analysis, and writing—original draft preparation: J.L.; software, visualization, and validation: X.L., L.Z. and X.Z.; writing—review and editing, X.W.; project administration, funding acquisition, resources, and supervision: Y.J. All authors have read and agreed to the published version of the manuscript.

**Funding:** This research was funded by the National Natural Science Foundation of China, grant number 51965059; Xinjiang Academy of Agricultural Sciences' Key Cultivation Project of Scientific and Technological Innovation, grant number xjkcpy—2021003; The Special Project of Basic Scientific Activities for Non-profit Institutes supported by the government of Xinjiang Uyghur Autonomous Region, grant number KY2022018.

**Institutional Review Board Statement:** Not applicable.

**Informed Consent Statement:** Not applicable.

**Data Availability Statement:** Not applicable.

**Conflicts of Interest:** The authors declare no conflict of interest.

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
