# Peer review of "Research on the Adaptability of High-Performance Film for Full Recycling to the Curl-Up Film Collecting Method"

_agriculture, doi:10.3390/agriculture12071051_

Round 1

Reviewer 1 Report

The article “Research on the adaptability of high-performance film for full recycling to the curl-up film collecting method” is devoted to the study of the strength characteristics of plastic film used for curl-up film collecting. Taking into account the relatively low tensile strength of existing films and rather large losses of time and labor that occur when the film breaks, the problem of studying the strength of such films is relevant.

The paper contains results of comprehensive studies, including a multifactorial experiment. In particular, the strength of specimens cut from films of different thicknesses and different types in two directions was evaluated to take into account the anisotropy of mechanical properties. The degradation of the mechanical properties of the studied films after various laying periods was also evaluated.

Among the paper advantages, it should be noted the full-scale experiment using different curl-up film collectors. The paper is of practical value: the obtained values of the films strength and plasticity characteristics under various curl-up conditions will make it possible to better adjust the equipment parameters and increase the efficiency of the collection process. The results obtained are not universal, but they provide a general understanding of the quantitative effect of various factors on the film mechanical properties.

It is necessary to note the following wishes for the paper:

1. Figure 5 is very difficult to analyze. The inscriptions in the figure are impossible to read, and the abundance of curves makes analysis too difficult.

2. Sometimes the article contains overly detailed descriptions of obvious facts. For example, in lines 279-288, a whole paragraph is devoted to describing a simple fact that was clear even before writing the paper (the need for uniform movement of the machine). Perhaps such descriptions should be given more concisely. This also applies to conclusions (Section 5).

3. The heading of table 3 needs to be corrected. Tables 3 also contains parameters that are not explained in the text (k1, k2, etc.).

The paper as a whole is well formed and easy to read.

Reviewer 2 Report

in this study contrast test was carried out on the tensile property of the high-performance film for full recycling and the ordinary PE film that is extensively applied in China. The study is innovative and can solve the problem of poor tensile performance of plastic film used in China and the difficulty in curl-up film collecting. However there is handling problem in the manuscript, try to sum up all the results in one chapter, introduction and some figures must be improved.
